# Assessment of Diagnostic Specificity of Anti-SARS-CoV-2 Antibody Tests and Their Application for Monitoring of Seroconversion and Stability of Antiviral Antibody Response in Healthcare Workers in Moscow

**DOI:** 10.3390/microorganisms10020429

**Published:** 2022-02-12

**Authors:** Vera S. Kichatova, Fedor A. Asadi Mobarkhan, Ilya A. Potemkin, Sergey P. Zlobin, Oksana M. Perfilieva, Vladimir T. Valuev-Elliston, Alexander V. Ivanov, Sergey A. Solonin, Mikhail A. Godkov, Maria G. Belikova, Mikhail I. Mikhailov, Karen K. Kyuregyan

**Affiliations:** 1Clinic & Research Institute for Molecular and Personalized Medicine, Russian Medical Academy of Continuous Professional Education, 125993 Moscow, Russia; axi0ma@mail.ru (I.A.P.); spzlobin@yandex.ru (S.P.Z.); operfileva@mail.ru (O.M.P.); michmich2@yandex.ru (M.I.M.); karen-kyuregyan@yandex.ru (K.K.K.); 2Laboratory of Viral Hepatitis, Mechnikov Research Institute for Vaccines and Sera, 105064 Moscow, Russia; 1amfa@bk.ru; 3Center for Precision Genome Editing and Genetic Technologies for Biomedicine, Engelhardt Institute of Molecular Biology, Russian Academy of Sciences, 119991 Moscow, Russia; gansfaust@mail.ru (V.T.V.-E.); aivanov@yandex.ru (A.V.I.); mariabelikova60@yandex.ru (M.G.B.); 4N.V. Sklifosovsky Research Institute for Emergency Medicine of the Moscow Health Department, 129090 Moscow, Russia; solonin@yahoo.com (S.A.S.); mgodkov@yandex.ru (M.A.G.); 5Laboratory of Molecular Pathogenesis of Chronic Viral Infections, NF Gamaleja Research Center of Epidemiology and Microbiology, 123098 Moscow, Russia; 6Scientific and Educational Resource Center for High-Performance Methods of Genomic Analysis, Peoples’ Friendship University of Russia (RUDN University), 117198 Moscow, Russia

**Keywords:** SARS-CoV-2, COVID-19, laboratory diagnosis, antibodies, healthcare workers

## Abstract

Anti-SARS-CoV-2 antibody testing is an efficient tool to assess the proportion of seropositive population due to infection and/or vaccination. Numerous test systems utilizing various antigen composition(s) are routinely used for detection and quantitation of anti-SARS-CoV-2 antibodies. We determined their diagnostic specificity using archived true-negative samples collected before the onset of the COVID-19 pandemic. Using test systems demonstrating 98.5–100% specificity, we assessed the dynamics of SARS-CoV-2 seroconversion and durability of anti-spike (S) antibodies in healthcare professionals (*n* = 100) working in Moscow during the first two cycles of the pandemic (May 2020 to June 2021) outside of the “red zone”. Analysis revealed a rapid increase in anti-SARS-CoV-2 seropositivity from 19 to 80% (19/100 and 80/100, respectively) due to virus exposition/infection; only 16.3% of seroconversion cases (13/80) were due to vaccination, but not the virus exposure, although massive COVID-19 vaccination of healthcare workers was performed beginning in December 2020. In total, 12.7% (8/63) remained positive for anti-SARS-CoV-2 IgM for >6 months, indicating unsuitability of IgM for identification of newly infected individuals. All except one remained seropositive for anti-S antibodies for >9 months on average. Significant (>15%) declines in anti-SARS-CoV-2 antibody concentrations were observed in only 18% of individuals (9/50). Our data on the high seropositivity rate and stability of anti-SARS-CoV-2 antibody levels in healthcare personnel working outside of the “red zone” indicate their regular exposition to SARS-CoV-2/an increased risk of infection, while a low frequency of vaccine-induced antibody response acquired after the start of vaccination points to vaccine hesitancy.

## 1. Introduction

The COVID-19 pandemic caused by SARS-CoV-2 is a significant threat for global health. Russia is among the countries with the highest number of registered COVID-19 cases [1]. The incidence of COVID-19 in Russia was sporadic until the end of March 2020. The first maximum in incidence rates (the so-called “first wave of the pandemic”) was reached by the first half of May 2020. The highest numbers of COVID-19 cases were then reported in the two largest metropolitan areas of the country, Moscow and St. Petersburg, comprising 19.9% and 8.6% of all infection cases in Russia, respectively [2].

The main method of laboratory diagnosis of COVID-19 is the determination of SARS-CoV-2 RNA and antigens [3,4], while the detection of antibodies to SARS-CoV-2 (anti-SARS-CoV-2) is being used for public health surveillance and epidemiologic purposes [5]. 

The serological window period for COVID-19 lasts for 2 to 3 weeks, making anti-SARS-CoV-2 antibody testing inapplicable for an early diagnosis of acute infection [6,7]. An exception may be anti-SARS-CoV-2 IgA, which can be detected within the first week after the infection [8]. Moreover, certain cohorts demonstrate low (<70%) rates of seroconversion; over 30% patients may remain seronegative for SARS-CoV-2 antibodies in multiple tests [9]. Due to these facts, neither the World Health Organization (WHO) nor the US Centers for Disease Control and Prevention (CDC) interim guidelines recommend the use of anti-SARS-CoV-2 antibodies as a standalone diagnostic tool [5,10].

The main serodiagnostic targets are antibodies to two viral proteins–spike (S) and nucleocapsid (N) [11,12,13,14]. Strong total antibodies, as well as single Ig classes IgA, IgM and IgG responses were reported in both severe patients and non-severe patients independently of the SARS-CoV-2 antigens used for the immunoassay: N or S proteins, S1 subunits or S protein receptor-binding domain (RBD) [15].

Despite the limited diagnostic value of anti-SARS-CoV-2 antibody testing, the Interim Guidelines of the Russian Ministry of Health state that presence of anti-SARS-CoV-2 antibody subclasses (IgA/IgM/IgG) can be used to discriminate different phases of the infectious process (acute disease vs. convalescence) [3].

The protective concentrations of antibodies are unlikely to be established due to emerging virus variants. Furthermore, some studies question the diagnostic value of the quantity of anti-SARS-CoV-2 antibodies, indicating that anti-spike antibody levels are not fully predictive of sterilizing immunity [16]. In view of these data, the lately interim recommendations of the US CDC state that serological tests should not be used to assess immunity after vaccination or to assess the need for vaccination of a previously unvaccinated person [5].

While having a limited role in disease diagnosis and still not a fully defined role in vaccine protection, anti-SARS-CoV-2 antibody testing represents an essential tool for assessing the proportion of the population who is seropositive, both due to the vaccination and virus exposure. Since a vast majority of available vaccines against COVID-19 are based solely on S protein (except for inactivated vaccines) [17], seroprevalence studies based on parallel testing for antibodies to N and S proteins can distinguish immunity resulting from infection and vaccination, as well as provide data on the duration of antibody response to different viral antigens.

Healthcare workers are highly exposed to SARS-CoV-2 and in need of the regular control of their health status, both for self-protection and for prevention of viral transmission to their patients. A number of studies characterized seroprevalence in this population category at the first wave of epidemics with rates varying from 2–3% in a region of South Denmark [18] to 30–45% in London [19,20].

According to the guidelines of the Russian Ministry of Health, testing for total or IgG anti-SARS-CoV-2 antibodies is recommended weekly for all healthcare workers (in parallel to RNA testing) who have not been previously tested or had a negative result in previous serological tests. Antibody testing discontinues after the detection of anti-SARS-CoV-2 resulting from either preceding infection or vaccination [21]. Such testing, as a part of monitoring of seroprevalence to SARS-CoV-2, requires sensitive, specific and standardized diagnostic test systems. As SARS-CoV-2 belongs to the genus betacoronavirus [22], some of its antigens that are used in serological tests may be similar to other human coronaviruses, causing potential false-positive results. Earlier studies have shown that out of six commercial SARS-CoV-2 enzyme linked immunoassays, only four provided >95% specificity [23]. Cross-reactivity was described of antibodies to S proteins of SARS-CoV-2 and four other betacoronaviruses, including SARS-CoV, MERS-CoV and seasonal coronaviruses OC43 and HKU1 [24,25]. In Russia, more than 100 diagnostic test systems for qualitative detection of anti-SARS-CoV-2 IgA, IgM and IgG antibodies (separately and in combination) have been registered, as well as six test systems for the quantitation of total antibodies or IgG [26]. The analytical and performance characteristics of these tests can vary significantly, which, in turn, can lead to a decrease in the reliability and reproducibility of the results of seroprevalence studies. Reliable seroprevalence assessment relies on a preceding rigorous assessment of the specificity of diagnostic test systems to be used for the detection of antibodies to S and N proteins.

The aim of this study was to determine the diagnostic specificity of the test systems employed for the detection and quantitation of antibodies to SARS-CoV-2, and by using the most specific test systems, assess the frequency of seroconversion and stability of anti-SARS-CoV-2 antibody response in a cohort of healthcare professionals from Moscow, working outside of the red zone, during the first two cycles of the epidemic process (May 2020 to June 2021).

## 2. Materials and Methods

### 2.1. Study Design

Design of this serological prospective observational study was approved by the Ethics Committee of the Russian Medical Academy of Continuous Professional Education in Moscow, Russia (Approval no. 6 dated 24 April 2020).

The study was performed in two steps, and each included several tasks (Figure 1). First, we assessed the specificity and limit of detection (LoD) of anti-SARS-CoV-2 test systems (hereafter referred as tests) that were widely used in clinical practice in Russia and had a different antigen composition (able to detect anti-N or anti-S antibodies). Further, we evaluated the correlation between cut-off indexes (COIs) obtained in qualitative tests and the anti-SARS-CoV-2 antibody concentrations obtained for the same samples in quantitative tests.

The second step of our study was the monitoring of the SARS-CoV-2 seroconversion using several evaluated serological tests in the cohort of healthcare professionals during the two first waves of COVID-19 in Moscow, Russia. This step included the assessment of seroconversion rates and the analysis of the dynamic changes in anti-SARS-CoV-2 concentration during the observation period (from 29 May 2020 to 30 June 2021), and the evaluation of the duration of anti-SARS-CoV-2 IgM antibodies’ reactivity in those who were IgM positive at the initial screening. All steps of the study and tests used are described in detail in the following subsections.

### 2.2. Study Participants and Study Samples

Diagnostic specificity of tests (Step1) were assessed by testing archive serum samples obtained from healthcare workers in the Belgorod region of Russia (*n* = 194) in 2018 and healthcare workers in the Kaliningrad region of Russia *(n* = 87) in January–August 2019, before the COVID-19 pandemic.

At the second step of the study, a total of 527 participants were screened for anti-SARS-CoV-2 IgM and IgG antibodies. Study participants were employees of the Russian Medical Academy of Continuous Education (RMANPO, Moscow, Russia) and included medical personnel (except workers out of the red zone), teachers, students, technical and administrative staff. No inclusion/exclusion criteria for age and sex were applied. Employees were repeatedly invited to undergo serology testing during the period from 29 May 2020 to 30 June 2021. All were eligible regardless of job or whether they had experienced a COVID-19–like illness. At each blood draw, study participants gave written informed consent and completed a questionnaire with demographic data and information on symptoms of acute respiratory infections, diagnosed SARS-CoV-2 infection and COVID-19 vaccination. Cohort/sub-cohorts of study participants are presented in Appendix A.

Sampling and serological testing was conducted according to the principles expressed in the Declaration of Helsinki. Written informed consent was obtained from all study participants. Serum samples were coded and stored in aliquots at −20 °C and −70 °C until testing (for serum 2020–2021 and 2018–2019, respectively). After thawing, sera were tested for anti-SARS-CoV-2 antibodies.

### 2.3. Tests Used to Assess Anti-SARS-CoV-2 Antibodies

Seven commercial serological tests certified for diagnostic use in the Russian Federation were used in the study, from both domestic and international manufacturers. Tests have different antigenic composition and are designed to detect different isotypes of immunoglobulins (Table 1). Anti-SARS-CoV-2 antibody testing and the interpretation of the results were carried out according to the instructions of the manufacturers of the respective tests.

### 2.4. Assessment of the Specificity of Anti-SARS-CoV-2 Tests

Diagnostic specificity of tests #1 (DS IFA-ANTI-SARS-CoV-2 versions 1, 2 and 4/5), #2 (SARS-CoV-2-IgG-IFA), #3 (SARS-CoV-2-IgG-IFA-BEST), #4 (SARS-CoV-2-IgM-IFA-BEST) and #5 (Elecsys Anti-SARS-CoV-2) were assessed by testing two panels of true negative archive serum samples (Appendix A). Each sample from both panels was tested in all five tests at least once. Early versions 1 and 2 of the test system #1 were only used for testing of the panel of archive samples from Belgorod region (*n* = 194). Specificity rates were calculated for each test as a percentage of pre-pandemic samples correctly identified as negative with 95% confidence interval (95% CI).

### 2.5. Determination of the Limit of Detection of Anti-SARS-CoV-2 Tests

The limit of detection (LoD) was calculated for test #1 (DS IFA-ANTI-SARS-CoV-2, version 4/5), test #2 (SARS-CoV-2-IgG-IFA) and test #3 (SARS-CoV-2-IgG-IFA-BEST) based on the testing of a dilution series of research reagent for anti-SARS-CoV-2 antibodies (NIBSC code 20/130) from the National Institute for Biological Standards and Control, UK. This reagent was shown to contain anti-SARS-CoV-2 antibodies at a concentration of 557 binding antibody units per mL (BAU/mL) in the study for the establishment of the WHO international standard and reference panel for anti-SARS-CoV-2 antibody [27].

For each test, the dilution series were obtained using sample diluent from the respective test. Based on the results of preliminary tests, the dilution series included dilutions 1:500, 1:1000, 1:2000, 1:4000 and 1:6000 for the DS IFA-ANTI-SARS-CoV-2 test and SARS-CoV-2-IgG-IFA test; 1:500, 1:1000, 1:1200, 1:1400 and 1:1600 for the SARS-CoV-2-IgG-IFA-BEST test. All dilution levels were tested in each test at least thrice. The endpoint titer was calculated using GraphPad Prism 9.0.2 (GraphPad Software, San Diego, CA, USA) for each test, and the average value was transformed into binding antibody units (BAU/mL).

### 2.6. Evaluation of Correlation between Cut-Off Indexes Obtained in Qualitative and Anti-SARS-CoV-2 Antibody Concentrations Obtained in Quantitative Test

COIs data were generated for 160 anti-SARS-CoV-2 reactive sera obtained from 34 prospectively followed, 4 sub-cohort of healthcare workers (Figure 1). Each participant was represented by 2 to 8 samples (M = 4.7; SD = 2.02) with ≥5 months between the first and last samplings (M = 6.76; SD = 1.43). 

Serum panel was analyzed first in the qualitative tests #3 (SARS-CoV-2-IgG-IFA-BEST) and #5 (Elecsys Anti-SARS-CoV-2), and then in the quantitative test #6 (Elecsys Anti-SARS-CoV-2 Sb) for anti-SARS-CoV-2 concentrations. All samples with anti-SARS-CoV-2 concentrations above the quantitation limit of test #6 (>250 BAU/mL) were diluted 100-fold with sample diluent from the kit manufacturer (Diluent Universal, Roche Diagnostics, Basel, Switzerland) and tested. The obtained result was multiplied for dilution factor to obtain the final concentration. Scatter diagrams were built, and Spearman and Kendall correlation coefficients between anti-SARS-CoV-2 concentrations were calculated for the paired data sets.

### 2.7. Monitoring of the Anti-SARS-CoV-2 Seroconversion in Cohort of Healthcare Workers

A total of 527 participants (2,101 serum samples, 2 to 22 per participant) were screened for anti-SARS-CoV-2 IgM and IgG antibodies from 29 May 2020 to 30 June 2021. Data from 527 blood sera obtained during the initial screening with test #3 were used to assess the baseline prevalence of anti-SARS-CoV-2 IgG in the study cohort (Appendix A, Task 2.1).

Among the 527 participants initially tested, further analysis involved only 120 employees who were available for long-term follow-up testing (Figure 2) and were further divided into sub-cohorts who met the following criteria, depending on the task of the study: 1) seropositive participants followed up for at least 4 month (*n* = 63, Appendix A, Task 2.2); 2) employees who had the first test conducted before June 2020 and whose anti-SARS-CoV-2 antibody status was known by the end of the study (June 2021; *n* = 100 Appendix A, Task 2.3) and 3) seropositive participants followed up for at least 6 months (*n* = 56, Appendix A, Task 2.4).

Among the 100 permanent employees of RMANPO who were prospectively followed throughout 13 months of the observation period, the SARS-CoV-2 seroconversion rate was calculated based on results of IgG antibodies’ testing using test #3 (SARS-CoV-2-IgG-IFA-BEST). In this sub-cohort, the mean number of visits was 9.86 (SD = 4.96). In those from this sub-cohort who had detectable anti-SARS-CoV-2 antibodies ≥6 months (*n* = 56), the antibody concentrations were measured using quantitative test #6 (Elecsys Anti-SARS-Cov-2 S) in each sample reactive in screening test #3.

Among study participants who were IgM antibodies’ reactive in initial testing and who were available for follow-up ≥ 4 months (*n* = 63), the duration of IgM detection was assessed based on repeated testing in test #4 (SARS-CoV-2-IgM-IFA-BEST) with confirmation of reactive results in test #7 (Mindray CLIA IgM).

### 2.8. Statistical Analysis

Data analysis was performed using graphpad.com (access date 14 November 2021). Statistical analysis included the calculation of a 95% confidence interval (95% CI) and assessment of the significance of differences in values between groups using Fisher’s exact test (significance threshold *p* < 0.05). Average values (M) and standard deviation (SD) were calculated using the Descriptive Statistics Analysis Package in Microsoft Office Excel. The correlation analysis included the calculation of Spearman and Kendall coefficients using IBM SPSS Statistics 28.0.0.0 (IBM Corp, Chicago, IL, USA, 2021).

## 3. Results

### 3.1. Performance of Anti-SARS-CoV-2 Tests

Among the studied seroassays, only test #5 (Elecsys Anti-SARS-CoV-2) yielded no false-positive results when testing the panel of pre-pandemic sera, gaining the specificity rate of 100% (95% CI: 98.4–100%). All other tests gave reactive results in several samples each, with mean COIs of reactive samples being within the range 1.80–2.61, depending on the test. The detailed data on diagnostic specificity and sensitivity of studied anti-SARS-CoV-2 tests are presented in Table 2.

The increase in specificity from 96.9% to 99.3% was observed for different versions of test #1 (DS IFA-ANTI-SARS-CoV-2). In early test versions 1 and 2, 62.5% (5/8) of false-positive results were observed in the same archive serum samples, while in test version 4/5, a false-positive result was obtained in serum samples negative in the test earlier versions (Appendix A). Excluding earlier versions of test #1, all tests had specificity ≥ 98.5%. When non-specifically reactive samples in tests of different manufacturers were analyzed, only one false-positive sample was reactive in more than one test; all other sera were reactive in only one test which confirmed unspecific nature of the reactivity (Appendix A).

The limit of detection was determined by using the dilution of a reference specimen (NIBSC code 20/130) for three tests: #1 version 4/5 (DS IFA-ANTI-SARS-CoV-2) which detect total antibodies both to N and S viral proteins; #2 (SARS-CoV-2-IgG-IFA) detecting IgG antibodies to RBD domain of S protein; and #3 (SARS-CoV-2-IgG-IFA-BEST) intended to detect IgG antibodies to S protein. The limit of detection varied between 0.1 to 0.5 BAU, with #1 generation 4/5 (DS IFA-ANTI-SARS-CoV-2) having the lowest limit of detection of 0.1 BAU/mL (Table 2, Appendix A).

Based on data obtained on tests performance and their pricing, test #4 was chosen for routine anti-SARS-CoV-2 antibody screening of employees of RMANPO.

### 3.2. Correlation of COIs Obtained in Qualitative #3 (SARS-CoV-2-IgG-IFA-BEST) and #5 (Elecsys Anti-SARS-CoV-2) Tests and Anti-SARS-CoV-2 Antibody Concentrations in Quantitative # 6 (Elecsys Anti-SARS-CoV-2 S) Test

The correlation between of COI values obtained in two qualitative tests based on different viral antigens (#3 for detection of anti-S IgG antibodies and #5 for detection of total anti-N antibodies; Table 1) and anti-SARS-CoV-2 concentrations measured in the quantitative test and expressed in BAU/mL (#6) was analyzed using 160 blood serum samples obtained from 34 participants who had time interval between the first and the last sampling ≥ 5 moths (Appendix A, Task 1.3). The linearity between COI values obtained in the test of different antigen composition and between COI values and concentrations in BAU/mL observed on scatter plots was weak, although Spearman coefficient was above 0.7 for correlation between COI values in anti-N and anti-S qualitative tests (Figure 3A, *p* < 0.01) and between COI values in anti-S test and antibody concentrations measured in quantitative test (Figure 3B, *p* < 0.01). Besides the strong scattering observed for all three comparisons, it should be noted that high concentrations in test #6 (>250 BAU/mL) were observed in some samples with rather low COI values: below 7.6 in test #3 (anti-S IgG, Figure 3B) and below 6.6 in test #5 (total anti-N, Figure 3C).

### 3.3. Anti-SARS-CoV-2 Prevalence Rates at Initial Screening Conducted at Different Stages of COVID-19 Pandemic

The routine seromonitoring of the employees of RMANPO was performed using test #3 SARS-CoV-2-IgG-IFA-BEST. The prevalence of anti-SARS-CoV-2 IgG at initial screening performed after the decline of the so-called “first wave” of the COVID-19 pandemic in Moscow (June–September 2020) was 17.8% (36/202). Anti-SARS-CoV-2 antibody monitoring of employees was expanded as the pandemic evolved. The seroprevalence in the employees recruited into screening at the later stages significantly increased during the second wave of the pandemic (October 2020–January 2021) constituting 27.2% (55/202, *p* = 0, 0317, Fisher’s exact test). It peaked to 52.8% (65/123, *p* > 0.0001, Fisher’s exact test) in participants who were tested for the first time during the decline of the “second wave” of pandemic in February–June 2021 (Figure 4).

### 3.4. Duration of Reactivity for IgM Antibodies to SARS-CoV-2

Durability of the anti-SARS-CoV-2 IgM antibody response was assessed among 63 participants who had ≥4 months between the first positive IgM test (not from the self-reported date of infection) and the last testing (126 to 396 days, M = 228.8; SD = 75.12) using test #4. Time interval between self-reported date of the disease and the first testing varied greatly and constituted <30 days in 61.9%, 30–60 days in 30.2% and >90 days in 7.9% participants. Participants with a sharp increase in the concentrations of anti-S total antibodies (described in Section 3.6) were excluded from this group to avoid the bias associated with the possible boosting of IgM production after repeated exposure to SARS-CoV-2.

Based on the data from test #4 that was used for routine IgM screening, 52.4% (33/63) participants were non-reactive for IgM antibodies at the initial testing or became consistently non-reactive after 2 to 3 months following the initial reactive test. At the same time, 31.7% (20/63) cases remained IgM antibody positive throughout observation period. In 12 participants, the duration of IgM reactivity ranged from 4 to 6 months (133–175 days); in eight participants, IgM antibodies were detected for more than 6 months (Figure 5A). Alternating positive, negative or grey zone results during the follow-up period were observed in 15.9% (10/63) participants (Figure 5B). Importantly, the data on the longevity of IgM antibody reactivity were determined only by the duration of the follow-up period, and did not reflect a loss of IgM reactivity with time.

Some of the late samples reactive in test #4 were tested in a second IgM test #7 to confirm the presence of anti-SARS-CoV-2 IgM antibodies. Only in 16.6% (3/18) of participants were reactive samples confirmed in the second IgM test, indicating the proven duration of reactivity for anti-SARS-CoV-2 IgM in these three participants >8 months (Figure 5). The last reactive samples from these three participants had COI values in test #7 of 9.5 (ID_425), 1.23 (ID_283) and 1.64 (ID_102).

### 3.5. Monthly Rates of Anti-SARS-CoV-2 Seroconversion in Cohort of Healthcare Professionals

Monthly anti-SARS-CoV-2 IgG seroconversion rates with dynamic changes in seroprevalence were assessed among 100 employees of the RMANPO who were screened on a regular basis (Figure 6). At the time of the first testing carried out at the decline of “first wave” of the COVID-19 pandemic in Moscow (early June 2020), 19% (19/100) of participants had detectable anti-SARS-CoV-2 IgG. By the end of June 2021, 80% (80/100) participants in the study cohort had detectable antibodies; however, only 16.3% (13/80) were vaccinated according to their self-report, while 83.7% (67/80) were not vaccinated by the time of this study and seroconverted due to the exposure to the virus. Among 67 naturally seroconverted participants, 76.1% (51/67) reported that they had been diagnosed with COVID-19 during the follow-up period, while 16.4% (11/67) noted that they did not have COVID-19 and 7.5% (5/67) chose not to answer this question in the questionnaire. In total, 9 out of these 16 seroconverted participants had anti-SARS-CoV-2 IgG before the start of vaccination campaign in Russia. We had conducted additional testing using test #5 for the other seven seroconverted participants: all of them had anti-N in their first anti-S reactive samples. Taken together, these data suggest that 16 participants with no reported history of COVID-19 were exposed to virus.

Two vaccinated participants (received different vaccines registered in Russia) in this cohort remained seronegative when tested for anti-S antibodies using test #3 and test #6 and were attributed to the group «IgG−» (Figure 6). 

Samples generating intermittent results in routine monitoring using test #3, such as gray zone or a negative result in a previously positive participant and vice versa, gray zone or a positive result in a participant who was negative in subsequent testing (*n* = 8), were retested on other tests. Except for three participants, false-positive or false-negative results were identified in all cases. Unusual serological profiles of these three participants, suggesting the loss of detectable antibodies and their subsequent appearance, were corroborated by independent tests using other tests (Appendix A). In two cases, antibodies remained only detectable in quantitative anti-S test #6 throughout the observation period (ID_49 and ID_187; Appendix A). In the third case, antibodies remained undetectable in all applied tests after the initially reactive sample until the 14th month of follow-up, when the participant became reactive again in all tests, including test #4 for detection of IgM antibodies (ID_79; Appendix A). This case is described in detail in Section 3.6 below. Participants with confirmed seroconversion, followed by seroreversion and repeated seroconversion were excluded from the analysis of seroprevalence (presented in Figure 6).

### 3.6. Changes in Anti-S Antibody Concentrations during the Follow-Up for More Than 6 Months

The quantitation of total anti-S antibodies using test #6 was carried out among 56 participants undergoing regular testing, who had 159 to 411 days after the first reactive test (M = 262, SD = 75.8). This sample did not include participants with post-vaccination antibodies.

According to the data reported by participants, 41.1% (23/56) had an asymptomatic infection and/or did not know about the previous COVID-19 infection, 25% (14/56) had a mild disease course (weakness, loss of smell, cough, headache and runny nose were reported as main symptoms) and 3.6% (2/56) suffered moderate or severe infection and were hospitalized. The remaining 30.3%, when filling out the questionnaire, skipped the column “presence of clinical manifestations of COVID-19”. Due to the limited sample size and the presence of an unspecified course of infection in 30.3% of cases, we did not analyze the dynamic changes in antibody concentrations depending on the clinical course of infection.

In 6 out of 56 cases, a sharp increase in anti-S antibody concentrations during the follow-up period was observed. Detailed anti-S antibody profiles for five participants with sharp increases in concentrations up to >250 BAU/mL, suggesting boosting due to the virus exposure, are shown in Appendix A.

Among the remaining 50 participants, no pronounced changes in the anti-S antibody concentrations were observed, which could have indicated viral exposition or infection, interfering with the assessment of the true longevity of the antibody response.

The distribution of peak anti-S antibody concentrations observed in participants during follow-up is shown in Figure 7A. In 41.1% (23/56) of cases, peak values were above the test’s upper level of quantitation (>250 BAU/mL), and in 21.4% (12/56) of cases they did not exceed 50 BAU/mL. For six participants presented in Appendix A, the peak values before the booster period were taken into account.

To identify the general dynamics of changes in antibody concentrations over time, a comparison was made between values obtained for each participant at first and the last testing. The difference between values <15% was not considered as significant.

In 56% (28/50) of the participants, anti-S antibody concentrations remained without significant changes within 7-13 months of follow-up (M = 9.4 months; SD = 2.4). In 26% (13/50), we registered a 1.2- to 8.5-fold (M = 2.5; SD = 2.0) increase in antibody concentrations during follow-up (7 to 14 months, M = 9.7 months; SD = 2.6). At the same time, in 18% (9/50) of participants, a 1.2- to 1.6-fold (M = 1.4; SD = 0.1) decrease was observed by 7-13 months of follow-up (M = 8.5 months; SD = 1.8) (Figure 7B).

## 4. Discussion

The humoral immune response to the SARS-CoV-2 proteins is believed to play a critical role in patients’ recovery, prevention of reinfection and post-vaccination protection. Therefore, in a short time after the start of the COVID-19 pandemic, one of the most important areas of research was the development of reliable serological tests for assessment of anti-SARS-CoV-2 antibodies of different subtypes and specificities, and the study of the kinetics of the induction, expansion and eventual contraction of antibody response. The first task of our study was to determine the diagnostic specificity and the limit of detection of the advanced and widely used tests for the routine diagnostics in Russia. Specificity rates obtained in our study on a panel of archived samples collected before the onset of the pandemic showed that all test systems (in their latest versions) exhibited specificity >98.5%. Rapid evolution with an increase in specificity after optimization was vividly illustrated on the example of versions 1, 2 and 4/5 of test system #1, when a more specific version replaced the previous, less specific ones.

The homology between amino acid sequence of the RBD region of SARS-CoV-2 S-protein and other widely and long-circulating human coronaviruses is within 19–21% [28]. N-protein is more conserved and shares a significant similarity between betacoronaviruses, including a highly conserved motif in its N-terminus and immunodominant epitope regions [29,30].

Nevertheless, anti-S and anti-N antibody tests in our study have shown no relation of the level of specificity to the nature of antigen utilized in the test, indicating that results of the tests were not confounded by cross-reactivity of sera to homologous proteins of other coronaviruses, and seroprevalence tests can be based on both S and N proteins.

At the same time, it is necessary to take into account the possibility of obtaining false-positive results due to nonspecific anti-N and/or anti-S antibody reactivity. Our study of the archived sera of healthcare workers collected prior to the pandemics in 2018 and January–August 2019 revealed that nonspecifically reactive samples were different in different tests, reflecting differences in the composition of their immunosorbents/antigens, and isotypes of detected antibodies. Hence, assessment of sera in multiple serological tests based on different antigens and antibody isotypes would help to exclude false positivity.

Another important characteristic of serological tests used for COVID-19 diagnostics is their detection limit; a high detection limit allows for not missing samples with low levels (low concentrations) of anti-SARS-CoV-2 antibodies. The introduction of the WHO International Standard for anti-SARS-CoV-2 antibody testing [27] greatly contributed to the harmonization of results of serological testing, thereby allowing, among other things, to determine the test’s limit of detection (LOD). Testing of the dilutions of reference sample (NIBSC code 20/130) traceable to WHO International Standard with three test systems used here for screening of seroprevalence, demonstrated that LOD between 0.1 to 0.5 BAU/mL; values were significantly lower than concentrations of anti-SARS-CoV-2 antibodies relevant for diagnosis of infection or vaccination. Thus, data on tests of high performance suggest that seroprevalence rates obtained in our study are not biased by the tests used.

Using these approbated tests, we characterized humoral response to SARS-CoV-2 in a cohort of employees of an organization that combines medical and educational activities in a metropolitan area, Moscow. Specifically, we determined (i) the duration of anti-SARS-CoV-2 IgM; (ii) the dynamics of anti-SARS-CoV-2 seroconversion during the first year of the pandemic in a metropolitan area of Moscow; and (iii) longevity of anti-S antibody response and fluctuations in anti-S antibody levels after infection.

Russian guidelines recommend a separate testing for antibodies of the IgM/IgA and IgG to diagnose COVID-19 and discriminate acutely infecting/seroconverting from convalescent patients [3]. Opposing this recommendation, our data suggest that up to 20% of initially IgM positive individuals may remain reactive within 4 to 6 months. It should be noted that upon receiving a positive result of detecting anti-SARS-CoV-2 IgM antibody, the study participants were sent for a RT-PCR testing, which gave a negative result. Thus, long-term reactivity for anti-SARS-CoV-2 IgM antibody was associated not with prolonged virus circulation, but with individual characteristics of the immune response. Our data corroborates conclusions made by Falzoe L et al. that anti-SARS-CoV-2 IgM antibody testing does not provide any advantage in terms of early diagnosis, since these immunoglobulins often appear simultaneously with IgG [6]. Moreover, it has been shown that anti-SARS-CoV-2 IgM antibody can be detected in persons who have been vaccinated against COVID-19 and have not been previously exposed to SARS-CoV-2 [31,32]. Taken together, these data indicate that anti-SARS-CoV-2 IgM antibody is not advisable to use for identification of newly infected individuals and should be excluded from the COVID-19 diagnostic system.

Due to limited diagnostic value of IgM antibodies, we performed seromonitoring in a studied cohort using IgG and total antibodies tests. We documented a rapid accumulation of immunity to SARS-CoV-2 in the observed cohort; the proportion of IgG seropositives at initial testing increased from 17.8% after the first epidemic surge in Moscow to 52.8% a year later, after the second wave of the epidemic. Similarly, in the subgroup of regularly tested participants, the proportion of seropositives quadrupled over the year, from 19% to 80%, with a small contribution of vaccination (only 13%), although the mass vaccination was started six months before the end of this study. Thus, these data reflect the general unfavorable epidemic situation in Moscow, which led to a high frequency of SARS-CoV-2 exposure among study participants. Noteworthy is the large proportion of seropositive individuals identified in the initial screening at the beginning of the study – 17.8%, which is significantly higher than the seroprevalence rates among medical workers observed in other countries after the first wave of the epidemic – only 0.85% in Poland [33], up to 7.2% in Turkey [34] and 10.3% in Spain [35]. According to a meta-analysis by Galanis et al. based on data obtained before 24 August 2020, i.e., following the first wave of the pandemic, SARS-CoV-2 seroprevalence among healthcare workers averaged at 8.7% with the highest rates observed in North America (12.7%) compared to Europe (8.5%), Africa (8.2%) and Asia (4%) [36]. However, in some studies, prevalence rates could be much higher even in Europe. For example, in a study conducted in May–June 2020 in the UK, seroprevalence among medical personnel was as high as 45% [20].

Some of these “first wave” studies reported seroprevalence in blood donors at 2%, in administrative personal of the hospitals and medical institutions at 3–6%, and up to 10% in medical staff not working in the red zone, an increase reflecting the degree of the probability of viral exposition. Illustrating this point, Lai et al. have shown that SARS-CoV-2 seroprevalence rates in healthcare personal properly using personal protective equipment do not exceed those among the general population [37].

Another important factor influencing the intensity of the SARS-CoV-2 spread among medical personnel may be non-compliance with the rules of social distancing in the workplace and, to a greater extent, in places of recreation [38].

It should be noted that the cohort followed up in this study did not include frontline medics working with COVID-19 patients, but included both doctors of other specialties and of the auxiliary and administrative employees. However, as has been previously shown in a number of epidemiological studies, non-frontline healthcare professionals and education workers may be at increased risk of SARS-CoV-2 infection due to the lower level of protection compared to the “red zone” alongside with high likelihood of encountering asymptomatic SARS-CoV-2 carriers [39,40,41].

Of note, in this study, the proportion of individuals who had asymptomatic infection and were unaware of having been infected, or at least indicated the absence of COVID-19 in the questionnaires, but were found to be seropositive, reached 16.4%. This proportion is consistent with data from sero-epidemiological studies, according to which asymptomatic infection is experienced by up to one third of infected persons [42].

The question of the durability of detectable anti-S antibody response, as well as the preservation of the neutralizing activity of these antibodies after the infection, remains open [43]. Monitoring the concentrations of post-infectious anti-S antibody among 50 study participants showed that in more than half of participants, they remained stable for 7–13 months, with over a 15% decrease in only 18% of the participants. In the longitudinally followed-up sub-cohort, we observed no disappearance of detectable anti-S antibody to SARS-CoV-2 except one participant, whose primary positive result could not be double-checked. Thus, detectable anti-S antibody may be retained for at least a year. This confirms observations from other studies, which demonstrated stable anti-SARS-CoV-2 antibody levels in the majority of subjects with previously diagnosed COVID-19 for 7 to 13 months after the infection [44,45,46,47,48,49]. Such stable levels of anti-SARS-CoV-2 antibodies may be attributed to repeated virus exposure among healthcare professionals during the pandemic. This is also evidenced by the high rate of seroconversion in the observed cohort.

The immunity developed after SARS-CoV-2 infection is not sterilizing, although it provides the short-term protection against re-infection in most cases [50]. The monitoring anti-SARS-CoV-2 antibody concentrations in the studied cohort revealed at least five cases of breakthrough infections, accompanied by a sharp rise in the concentration of anti-S antibodies.

The quantification of anti-SARS-CoV-2 antibodies is currently a highly sought-after test, although the protective level of antibodies is yet unknown, especially in the case of emerging virus variants. Thus, monitoring total antibody concentrations may be inappropriate for estimating the duration and degree of protection provided by COVID-19 vaccines. Still, sometimes attempts are made to monitor levels of antibodies using not quantitative, but qualitative tests based on COI values. Here, we have shown that there is no pronounced correlation between anti-S antibody concentrations and COI values obtained in qualitative tests, confirming that COI values cannot be used as a surrogate marker of antibody levels for diagnostic purpose.

In our cohort of healthcare workers, contribution of seroconversion due to vaccination was very small (only 13%). Such a low proportion was unexpected, since the campaign for COVID-19 vaccination of healthcare workers began on 5 December 2020, i.e., six months before the end of this study. Altogether, these data reflect the general unfavorable epidemic situation in Moscow, resulting in a high frequency of SARS-CoV-2 exposure among study participants, as well as a high level of vaccine hesitancy even among healthcare professionals. Noteworthy, a few months after the completion of this study, the strong recommendation for the vaccination among employees of the RMANPO was issued, resulting by the end of October 2021 in 66% (*n* = 66/100) coverage of the sub-cohort of healthcare workers longitudinally followed in this study. Information on the vaccination status of the remaining 34% of participants was unavailable.

Poor knowledge on vaccination and vaccine hesitancy among healthcare providers have been demonstrated previously in the context of other viral vaccine-preventable infections, such as HPV and HBV [51,52,53]. In case of SARS-CoV-2, such vaccine hesitancy among healthcare workers is associated with increased risk of infection not only for healthcare workers, but also for their patients, suggesting the need for legislative regulation of vaccination of this category of population, alongside a strict control over its implementation.

This study has several limitations. First, the participants in this study underwent voluntary periodic testing for anti-SARS-CoV-2 antibodies, but a significant number of them were tested once and were not available for repeated testing, since this test is not mandatory for the diagnosis of COVID-19 in accordance with international recommendations [10] and the Interim Guidelines for Prevention, Diagnosis and Treatment of COVID-19 published by the Russian Ministry of Health [3]. Only about 23% (120/557) of the participants were tested regularly, which is a definite limitation of this study. A second limitation is that when determining the presence of a previous SARS-CoV-2 infection or symptoms of an acute respiratory infection, we relied on the participants’ statements, which may be open for bias due to subjective perception, even though the majority of participants were medical professionals. Despite these limitations, our data objectively represent the current situation with anti-SARS-CoV-2 antibody testing in Russia and provide the data on prevalence and duration of postinfectious antibody response to SARS-CoV-2 proteins.

## 5. Conclusions

The data obtained indicate a high specificity of the tests used to detect anti-SARS-CoV-2 antibodies and the importance of confirming test specificity using panels of archived samples collected before the onset of the COVID-19 pandemic. We observed that a significant proportion of individuals retained anti-SARS-CoV-2 IgM antibodies for over 4–6 months, indicating their inapplicability for identification of newly infected individuals. We also established an early and rapid seroconversion for anti-SARS-CoV-2 antibodies among medical professionals due to virus exposure/infection achieved during the first year of the pandemic. Furthermore, our data demonstrated that the majority of infected individuals remained seropositive for antibodies to S protein of SARS-CoV-2 over a year, without significant decline in antibody levels, possibly due to repeated antigen exposures. The fact that in >70% of cases, seroconversion resulted from infection, not vaccination, points to a low level of vaccine acceptance among healthcare professionals, and calls for the necessity of legislational and educational counter-measures.

## Figures and Tables

**Figure 1 microorganisms-10-00429-f001:**
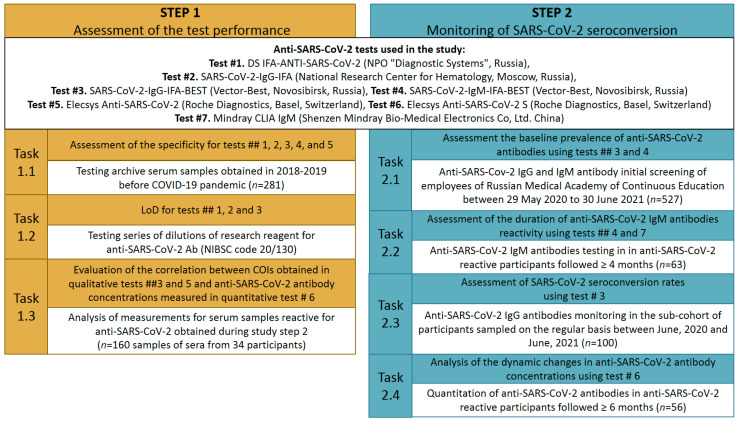
Study design. LOD, limit of detection. COI, cut-off index.

**Figure 2 microorganisms-10-00429-f002:**
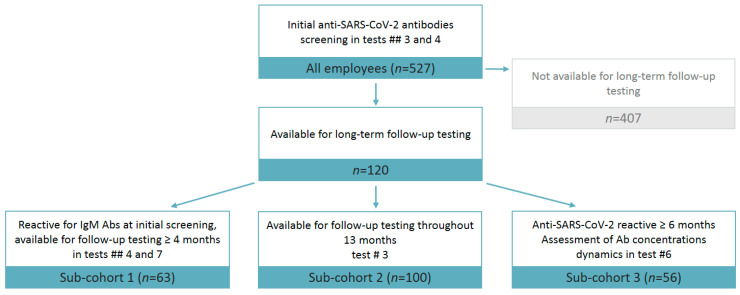
The distribution of study participants in sub-cohorts available for anti-SARS-CoV-2 antibody testing. Ab, Antibody.

**Figure 3 microorganisms-10-00429-f003:**
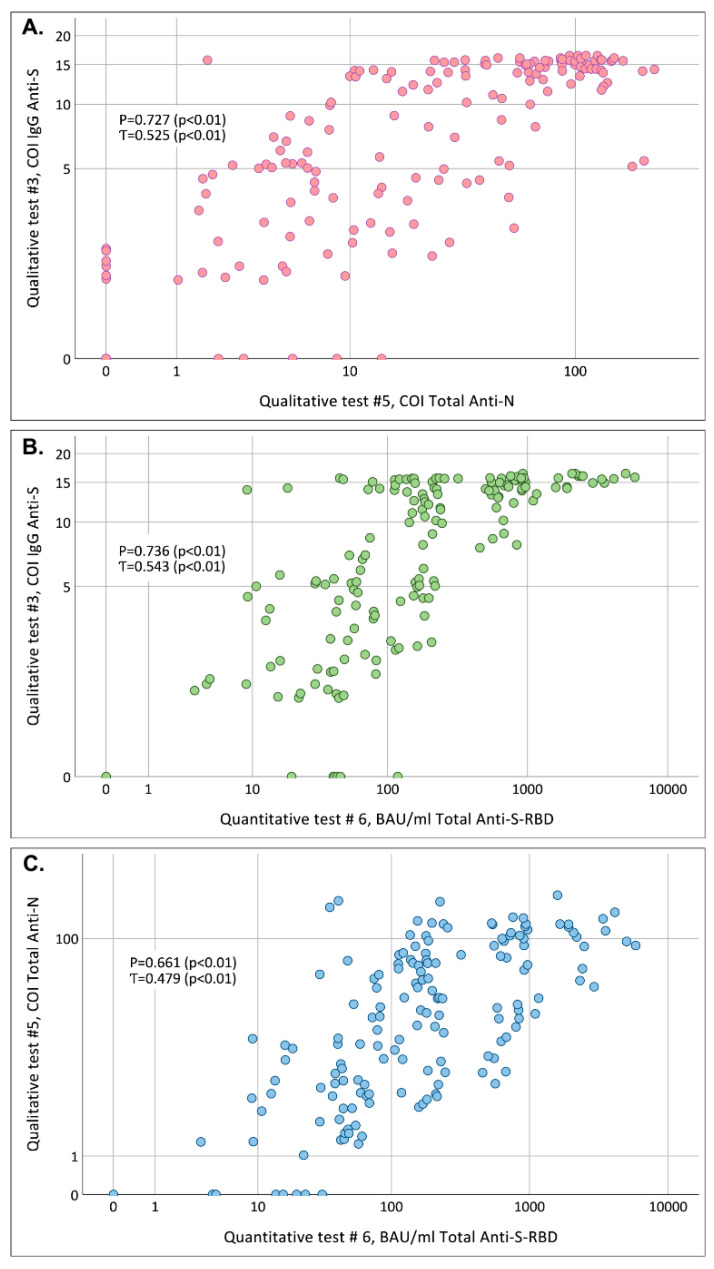
Scatter plots characterizing the correlation between (**A**) COIs obtained in test #3 (anti-S IgG) and test #5 (anti-N total antibodies), pink dots; (**B**) between COIs obtained in test #3 (anti-S IgG) and anti-SARS-CoV-2 concentrations measured in test #6 (total anti-S), green dots; (**C**) between COIs obtained in test #5 (total anti-N) and anti-SARS-CoV-2 concentrations measured in test #6 (total anti-S), blue dots; tested in the same panel of samples (*n* = 160). P: Spearman coefficient; Ƭ: Kendall coefficient. Note: when plotting the graphs, samples with negative results and in the gray zone were taken as “0”. COI, cut-off index. BAU/mL, binding antibody units per mL. N, nucleocapsid protein. S, spike protein. RBD, receptor binding domain.

**Figure 4 microorganisms-10-00429-f004:**
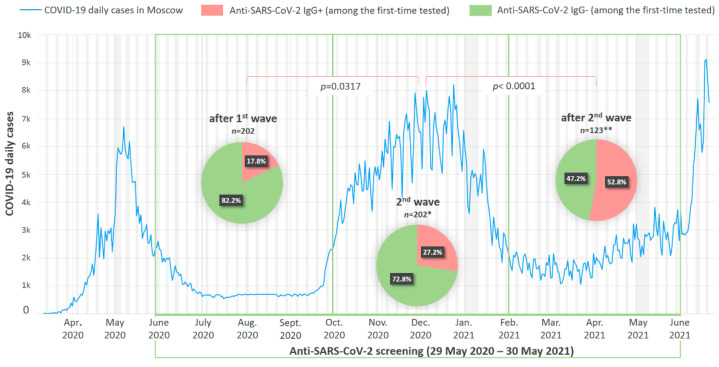
Anti-SARS-CoV-2 IgG antibody detection rates among first-time tested participants at different stages of pandemic. In the background, the official statistics of new daily cases (in thousands of cases) of COVID-19 in Moscow are presented [2]. At the bottom of the timeline, the duration of the study is marked in green. * the proportion of vaccinated among seropositive is unknown; ** the proportion of vaccinated among seropositive participants was 4.8% (6/123).

**Figure 5 microorganisms-10-00429-f005:**
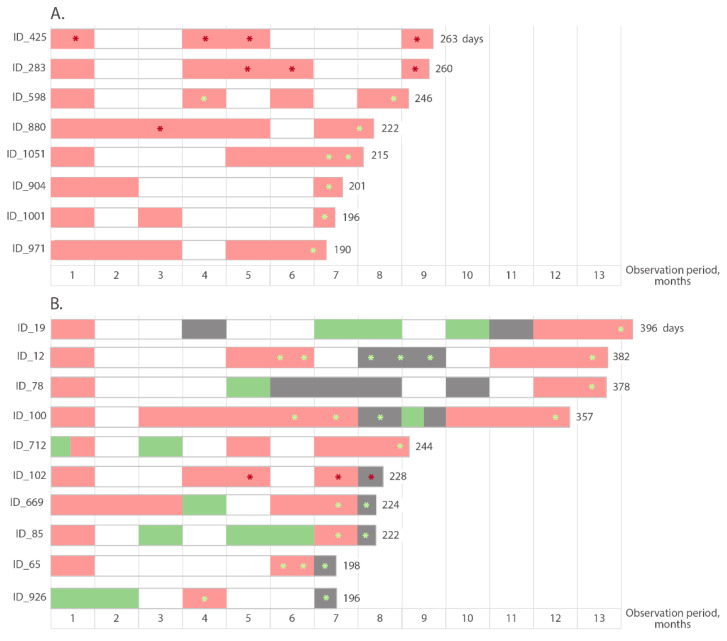
Schematic representation of the prolonged detection of SARS-CoV-2 IgM antibodies. (**A**) cases of continuous prolonged detection of IgM antibodies; (**B**) cases with alternating positive/negative/indeterminate (gray zone) results. Pink rectangles represent months with a positive result in test #4, green rectangles represent months with a negative result in test #4, gray rectangles represent an indefinite (gray zone) result in test #4. Red asterisks indicate positive results confirmed in the second IgM test #7, light green asterisks indicate negative results in test #7. The day of the last testing is indicated at the end of the columns.

**Figure 6 microorganisms-10-00429-f006:**
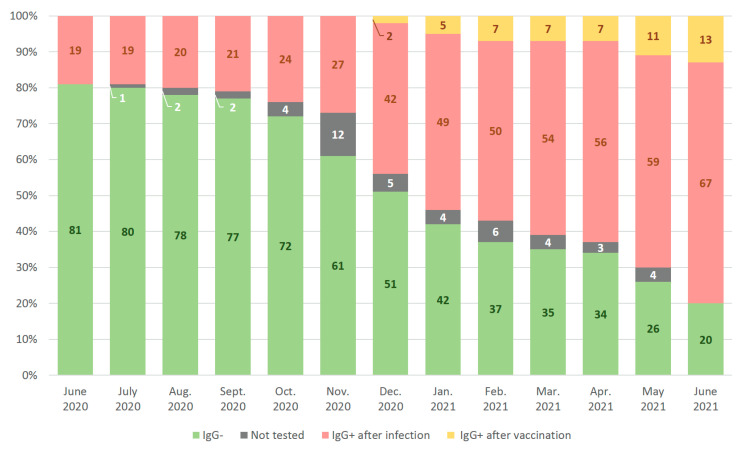
Monthly anti-SARS-CoV-2 IgG seroconversion rates among 100 participants monitored between June 2020 and June 2021. Note: Participants vaccinated against the background of the detectable post-exposure antibodies are still assigned to the «IgG + after infection» group on the graph. All seropositive participants who did not come to the testing in the following months continued to be counted as IgG +, if remained positive in the last test.

**Figure 7 microorganisms-10-00429-f007:**
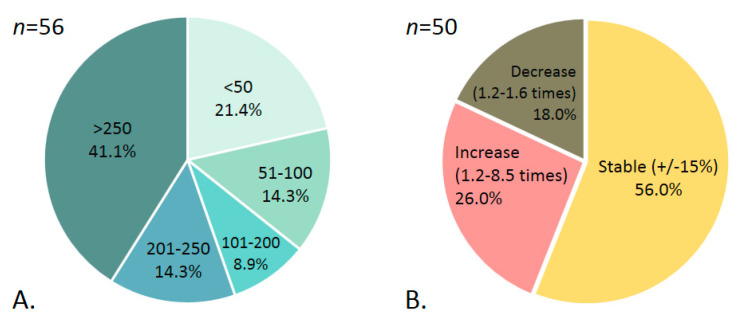
Distribution of (**A**) peak anti-S antibody concentrations observed in participants during follow-up ≥ 6 months (BAU/mL); (**B**) dynamic changes in anti-S concentrations throughout the follow-up.

**Table 1 microorganisms-10-00429-t001:** Test systems assessing anti-SARS-CoV-2 antibodies used in the study.

Test No.	Test System (Manufacturer)	Type of Test	Antibody Isotype	Target Protein	Positive Result	Grey Zone
1 *	DS IFA-ANTI-SARS-CoV-2 (NPO “Diagnostic Systems”, Russia)	Qualitative, ELISA	IgG + IgM	N and S	COI > Cut-off +20%	Yes (COI between Cut-off −20% to Cut-off +20%)
2	SARS-CoV-2-IgG-IFA (National Research Center for Hematology, Moscow, Russia)	Qualitative, ELISA	IgG	S-RBD	COI ≥ 1.1	Yes (COI from 0.9 to 1.1)
3	SARS-CoV-2-IgG-IFA-BEST (Vector-Best, Novosibirsk, Russia)	Qualitative, ELISA	IgG	S	COI ≥ 1.1	Yes (COI from 0.8 to 1.1)
4	SARS-CoV-2-IgM-IFA-BEST (Vector-Best, Novosibirsk, Russia)	Qualitative, ELISA	IgM	N and S-RBD	COI ≥ 1.1	Yes (COI from 0.8 to 1.1)
5	Elecsys Anti-SARS-CoV-2 (Roche Diagnostics, Basel, Switzerland)	Qualitative, CLIA	Total Abs	N	COI ≥ 1.0	No
6	Elecsys Anti-SARS-CoV-2 S (Roche Diagnostics, Basel, Switzerland)	Quantitative, CLIA	Total Abs	S-RBD	U/mL > 0.8	No
7	Mindray CLIA IgM (Shenzen Mindray Bio-Medical Electronics Co, Ltd. China)	Qualitative, CLIA	IgM	N and S	COI ≥ 1	No

* Test DS IFA-ANTI-SARS-CoV-2 specificity was assessed separately for versions 1, 2 and 4/5. ELISA, Enzyme-Linked Immunosorbent Assay. CLIA, Chemiluminescent immunoassay. N, nucleocapsid protein. S, spike protein. RBD, receptor binding domain. COI, cut-off index. The kits used for routine antibody screening in the study tasks 2.1, 2.2 and 2.3 (Figure 1) are highlighted in color.

**Table 2 microorganisms-10-00429-t002:** Specificity and limit of detection values of anti-SARS-CoV-2 tests.

Test No.	Test System (Manufacturer)		N Reactive Samples/ N Tested	Specificity of Test	Mean COI in Reactive Samples	N Samples in Grey Zone/ N Total	LoD, BAU/mL
1	DS IFA-ANTI-SARS-CoV-2 (NPO “Diagnostic Systems”, Russia)	version 1	6/194	96.9% (93.3–98.7%)	1.73	5/194	n.d.*
		version 2	7/194	96.4% (92.6–98.4%)	1.66	3/194	n.d.
		version 4/5	2/281	99.3% (97.3–99.9%)	2.79	3/281	0.1
2	SARS-CoV-2-IgG-IFA (National Research Center for Hematology, Moscow, Russia)		3/281	98.9% (96.8–99.8%)	2.61	3/281	0.4
3	SARS-CoV-2-IgG-IFA-BEST (Vector-Best, Novosibirsk, Russia)		4/281	98.5% (96.3–99.6%)	1.80	2/281	0.5
4	SARS-CoV-2-IgM-IFA-BEST (Vector-Best, Novosibirsk, Russia)		3/279	98,9% (97.3–99.9)	2.11	1/279	n.d.
5	Elecsys Anti-SARS-CoV-2 (Roche Diagnostics, Basal, Switzerland)		0/281	100% (98.4–100%)	-	0/281	n.d.

* n.d. = not determined. COI, cut-off index. LOD, limit of detection. BAU/mL, binding antibody units per mL.

## Data Availability

The data presented in this study are available in this article.

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
