# Peer review of "Assessment of Diagnostic Specificity of Anti-SARS-CoV-2 Antibody Tests and Their Application for Monitoring of Seroconversion and Stability of Antiviral Antibody Response in Healthcare Workers in Moscow"

_microorganisms, 2022, doi:10.3390/microorganisms10020429_

Round 1

Reviewer 1 Report

The authors intend to assess the diagnostic specify of anti-SARS-CoV-2 antibody tests and apply these tests for monitoring of seroconversion and stability of antiviral antibody response in healthcare workers in Moscow. The findings in this article are informative.

Several suggestions:

  1. Line 53, please add a reference after [SARS- CoV-2 RNA].
  2. Line 61, please change [Region-binding Domain (RBD)] to [Receptor-binding Domain (RBD)].
  3. Line 116, please change [SARS] to [SARS-CoV].
  4. Line 236, please delete the [T] after [Figure 2.].
  5. Line 261, why add [(98.4-100%)] after [100%]?
  6. Line 339, [Figure 5A] not [a]; line 340, [Figure 5B] not [b].
  7. Lines 367-370, why not confirm the cases [while 16,4% (11/67) noted that they did not have COVID-19 and 7,5% (5/67) chose not to answer this question in the questionnaire.] using test #5 to detect anti-N to know whether they are vaccinated or naturally infected?
  8. Line 387, is the [except 3] necessary?
  9. Line 461, please add a reference after [post-vaccination protection].
  10. Line 504, [LOD] or [LOI]?
  11. Line 519, [PCR] should be changed to [RT-PCR] or [real-time RT-PCR].
  12. The title in [Supplementary Figure 1 CAPTION] is [Supplementary Table S1. Specificity and limit of detection values of anti-SARS-CoV-2 tests.], please check: Table?, where is [limit of detection values]?

Reviewer 2 Report

The presented manuscript on „Assessment of diagnostic specify of anti-SARS-CoV-2 antibody tests and their application for monitoring of seroconversion and stability of antiviral antibody response in healthcare workers in Moscow“ provides information on the specificity of several systems for detection of anti-SARS-CoV-2 antibodies and some information on the seroconversion rate among healthcare workers in Moscow.

The manuscript needs to be substantially revised to reach readability and soundness for the readers.

Major:

  1. The title … the wording „Assessment of diagnostic specify of anti-SARS-CoV-2 antibody tests“ doesn’t make sense.
  2. The abstract …
    1. i) the herd immunity in SARS-CoV-2 was questionable for wuhan/british/delta variants and is likely not existent for omicron variant. thus the term „herd immunity“ should be avoided throughout the text. Anti-SARS-CoV-2 antibodies represent a marker of immunization by infection or vaccination;
    2. ii) the sentence „only 16.3% of seroconversion cases (13/80) were due to vaccination, although massive COVID-19 vaccination of healthcare workers was performed from December 2020“ may be interpreted in a way that majority of healthcare workers were vaccinated but did not develop antibody response. I assume that the majority of hospital personnel was just not willing to be vaccinated despite massive vaccination effort. Generally, the vaccine hesitancy and the need for its counteracting is not supported or related to the presented data and should be omitted for the text (last 4 lines in the abstract).
  3. The introduction
    1. Generally, it needs to be shortened and the information that are not to the point and are out-of-date currently or may be out-of-date or not valid in few months should be avoided. The past for should be used.
    2. g. “Russia, together with United States, India, Brazil and United Kingdom, is among countries with the highest number of registered COVID-19 cases” … may be omitted
    3. g. “According to official statistics (dated September 23, 2021), the number of cases of infection in Russia amounted to more than 7.3 million people with the mortality rate as high as 2.74% (201 445 cases per 7 354 47 995 infected)” … should be omitted as it is perishable information, not related to the merit of the manuscript and with the validity that may suffer from underreporting of SARS-CoV-2/covid-19 in Russia.
    4. “presence of anti-SARS-CoV-2 anti-body subclasses (IgA/IgM/IgG) can be used to discriminate different phases of the infectious process (acute disease vs. convalescence).” … this is not true. IgM and IgG occur almost simultaneously after infection and detectability of IgM suffers from low sensitivity depending on detection system and severity of clinical picture of SARS-CoV-2 infection on one hand, and on the other hand may be present weeks or months after the immunization (both infection or vaccination).
    5. “However, the serological window period for COVID-19 lasts for 2 to 3 weeks, making anti-SARS-CoV-2 antibody testing inapplicable for an early diagnosis of acute infection” … the serological window period depends again on the detection system and severity of clinical picture of SARS-CoV-2 infection. E.g. anti-nucleocapsid on Elecsys antibodies are detectable by 7 days from initial symptoms. More literature should be searched and text up-dated.
    6. “At the same time, according to the WHO report, the presence of post-infectious antibodies to SARS-CoV-2 is the best evidence of protection against re-infection for a period of at least 5-7 months, providing the protection in 81-100% of cases for persons under 65 years and in 47% for people over 65 years,” … likely not true now with omicron; should rephrased or omitted.
  4. Material and Methods
    1. Generally, it is difficult to grasp the study design as described using figure 1 with “steps” and “tasks” together with table 1. It may be helpfully to merge the information into one diagram and place it into supplementary material.
    2. It should may be specified who was the study population for seroconversion …the healthcare workers out of the red zone vs. “medical personnel, teachers, students, technical and administrative staff”.
  5. Results
    1. Specificity of the tests as one of the major observation and point of the article should be shown in the main text and discussed further
    2. Data on LoD detection are not shown
    3. Correlation of various tests (quantitative and qualitative) should not be performed in presented way. The concentration of SARS-CoV-2 Ab by quantitative anti-S test in BAU/ml should be measured in diluted samples (100x) to obtain the exact level when exceeding 250 BAU/ml.
    4. The case reports (figure and their description) from 3.6 should be moved to supplementary material.
    5. Figure 8. Does not provide much meaningful information. The time-dependent pattern of anti-SARS-CoV-2 Ab should be depicted in conventional way instead. The concentration of SARS-CoV-2 Ab by quantitative anti-S test in BAU/ml should be measured in diluted samples (100x) to obtain the exact level when exceeding 250 BAU/ml.
  6. Discussion
    1. Should be substantially shortened, strictly avoid perishable that are loosely related to the merit and should pertain the observations presented in the result section.

Reviewer 3 Report

Dear Reviewers,

The article “Assessment of diagnostic specify of anti-SARS-CoV-2 antibody tests and their application for monitoring of seroconversion and stability of antiviral antibody response in healthcare workers in Moscow” is interesting and deals with an important topic for the prevention of the COVID-19 pandemic. Some revisions are needed before it can be considered for publication, as per my comments here below.

Best regards,

The Reviewer

Abstract

  • Lines 20-21: Authors state “We determined their diagnostic specificity using archived samples collected before the onset of COVID-19 pandemic”. This can be not fully clear, especially considering the m,inited information available in the abstract: I would add “in identifying true negatives” after specificity.
  • The number of healthcare workers investigated has to be reported in the abstract.

Introduction

  • Lines 42-44: Ref 1 is not correct, as it refers to mortality. You should refer her to cumulative number of cases: https://covid19.who.int/ . And you should add also France, which is in the 5th position, just before Russia.
  • Lines 46-47: please update the data to January 2022
  • Lines 53-53: please change this sentence “The detection of antibodies to SARS-CoV-2 (anti-SARS-CoV-2) is considered as the surrogate marker of infection”. This is not correct. Antibodies presence may indicate an ongoing infection, a previous exposure/infection or a previous vaccination. Furthermore, their absence cannot be considered indicative of an absence of infection/previous exposure (or infection/vaccination. For these reasons anti-SARS-CoV-2 antibodies tracking is mainly useful for epidemiologic purposes and for the evaluation of a yes/no (not quantitative) response after vaccination.
  • Lines 54-55: the problems in considering antibodies testing as appropriate markers to monitor the infection has been identified quite soon. They were used also because diagnostic swab test for Rna detection were scarcely available at the beginning of the epidemic and not sufficient to monitor the spread of the infection.
  • Lines 56-59: from “Since” to “infection”: I should delete this sentence as it can be misleading for the previously reported problems in considering antibody testing as an appropriate test for the detection of infections in the case of SARS-CoV-2.
  • Actually you mention these problems in lines 75-77, but this results further misleading for the reader, as a few lines above it seems that you recommend antibodi3es testing as a diagnostic tool.
  • Lines 81-85_ unfortunately COVID-19 data becomes old quite soon, and this reported information could be true for the main virus variants circulating in June 2021, but I am afraid that this is not true now, particularly considering Omicron variant.
  • To avoid possible lack of clarity and contrasting messages for the readers, my suggestion is to reduce the length of the introduction, in particular considering lines 42-95. The message on the current internationally recognized role of antibody testing should be clear, while more “historical” data should be limited to a very few words.

  1. Materials and Methods

2.1. Study design

- It is not clear from your description and form the figure whether your study is prospective or retrospective, and neither whether it was observational or interventional.

2.2. Study participants and study samples

- What about inclusion/exclusion criteria for age and sex?

2.3. Tests used to assess anti-SARS-CoV-2 antibodies

- Table 2: the abbreviation “COI” has to be explained also in the table. Why there is for test 3 “COI≥1,1”?

- I would clearly indicate in the Table whether the tests are quantitative or qualitative, chemiluminescence or enzyme immunoassay tests.

2.4. Assessment of the specificity of anti-SARS-CoV-2 tests

- Lines 189-191 have information previously reported in paragraph 2.2

  1. Results

Congratulation for this section and for the figures: the analysis done are very detailed and provide important information.

  1. Discussion

- Lines 460-65: for the previously mentioned problems in the role of antibody testing I would delete these sentences, starting form “By October 2020…”

- Lines 490-496: when the latest 2019 samples were collected? Is there a difference between specificity in the testing of the 2018 and 2019 population? If there is a difference, and if the 2019 samples have been collected also in November-December 2019, a possible exposure to SARS-CoV-2 of these subjects cannot be fully excluded, even if unlikely.

- Line 481: “sequence the RBD”. I believe it should be “sequence of the RBD”

- It seems to me that you tested also IgA: what are the conclusions of your results considering these specific antibodies?

- Lines 526-28: why IgM only? I believe that also other types of Ig are not useful for diagnostic purposes

  1. Conclusions

- Please add “antibodies” after “anti-SARS-CoV-2”
